# Corticosteroid Resistance in Smokers—A Substudy Analysis of the CORTICO-COP Randomised Controlled Trial

**DOI:** 10.3390/jcm10122734

**Published:** 2021-06-21

**Authors:** Pradeesh Sivapalan, Andras Bikov, Charlotte Suppli Ulrik, Therese Sophie Lapperre, Alexander G. Mathioudakis, Mats Christian Højberg Lassen, Kristoffer Grundtvig Skaarup, Tor Biering-Sørensen, Jørgen Vestbo, Jens-Ulrik S. Jensen

**Affiliations:** 1Department of Medicine, Section of Respiratory Medicine, Herlev and Gentofte Hospital, University of Copenhagen, 2900 Hellerup, Denmark; jens.ulrik.jensen@regionh.dk; 2Department of Internal Medicine, Zealand University Hospital, 4000 Roskilde, Denmark; 3Wythenshawe Hospital, Manchester University NHS Foundation Trust, Manchester M23 9LT, UK; andras.bikov@gmail.com; 4Division of Infection, Immunity and Respiratory Medicine, University of Manchester, Manchester M13 9NT, UK; Alexander.Mathioudakis@manchester.ac.uk (A.G.M.); jorgen.vestbo@manchester.ac.uk (J.V.); 5Department of Respiratory Medicine, Copenhagen University Hospital-Hvidovre, Denmark and Institute of Clinical Medicine, University of Copenhagen, 2200 Copenhagen, Denmark; csulrik@dadlnet.dk; 6Department of Clinical Medicine, Faculty of Health and Medical Sciences, University of Copenhagen, 2200 Copenhagen, Denmark; 7Department of Respiratory Medicine, Antwerp University Hospital and Laboratory of Experimental Medicine and Pediatrics, University of Antwerp, 2000 Antwerp, Belgium; therese.lapperre@uza.be; 8The North West Lung Centre, Wythenshawe Hospital, Manchester University NHS Foundation Trust, Manchester M13 9WL, UK; 9Department of Cardiology, Herlev and Gentofte Hospital, University of Copenhagen, 2900 Hellerup, Denmark; mats.christian.hoejbjerg.lassen@regionh.dk (M.C.H.L.); kristoffer.grundtvig.skaarup@regionh.dk (K.G.S.); Tor.Biering-Soerensen@regionh.dk (T.B.-S.); 10Department of Biomedical Sciences, Faculty of Health and Medical Sciences, University of Copenhagen, 2200 Copenhagen, Denmark

**Keywords:** chronic obstructive pulmonary disease, corticosteroid resistance, smoking, airway inflammation, blood eosinophils

## Abstract

The CORTICO-COP trial showed that eosinophil-guided corticosteroid-sparing treatment for acute exacerbation of chronic obstructive pulmonary disease was non-inferior to standard of care and decreased the accumulated dose of systemic corticosteroids that patients were exposed to by approximately 60%. Smoking status has been shown to affect corticosteroid responsiveness. This post hoc analysis investigated whether eosinophil-guided treatment is non-inferior to conventional treatment in current smokers. The main analysis of current smokers showed no significant difference in the primary endpoint, days alive, and out of hospital within 14 days between the control group (mean, 9.8 days; 95% confidence interval (CI), 8.7–10.8) and the eosinophil-guided group (mean, 8.7 days; 95% CI, 7.5–9.9; *p* = 0.34). Secondary analyses of the number of exacerbations or deaths, the number of intensive care unit admissions or deaths, lung function improvement, and change in health-related quality of life also showed no significant differences between the two groups. The results of a sensitivity analysis of ex-smokers are consistent with the main analysis. Our results suggest that eosinophil-guided treatment is non-inferior to standard of care in current smokers and ex-smokers. Because data on the impact of smoking status on eosinophil-guided treatments are sparse, more randomised trials are needed to confirm our results.

## 1. Introduction

Systemic corticosteroids have anti-inflammatory effects and can alleviate symptoms in patients with moderate-to-severe chronic obstructive pulmonary disease (COPD) exacerbations [1,2,3,4]. However, corticosteroid treatment also leads to harmful side effects, including increased risk of infections, osteoporosis, venous thromboembolisms, and hyperglycaemia [5,6,7,8,9,10]. Therefore, several corticosteroid studies have investigated treatment regimens that may reduce the risk of harmful side effects while retaining the beneficial effects [11,12,13].

Blood eosinophils are surrogate markers for predicting the response to steroid treatment in patients with COPD [14]. The CORTICO-COP trial investigated a biomarker-guided approach to corticosteroid treatment based on daily blood eosinophil count [12]. In the trial, patients with blood eosinophil levels equal or above a threshold of 0.3 × 10^9^ cells/L received corticosteroid treatment, while treatment was withheld on treatment days when blood eosinophil levels were below that threshold. The main finding was that the eosinophil-guided strategy was non-inferior to standard treatment in terms of endpoints, and decreased the accumulated dose of systemic corticosteroids that patients were exposed to by approximately 60%. 

Smoking status was shown to precipitate corticosteroid responsiveness [15]. Smoking can increase oxidative stress in tissues, thereby suppressing the expression and activity of histone deacetylase-2, which is necessary for corticosteroids to exert their anti-inflammatory effects [15]. In addition, inhaled corticosteroids (ICS) are less effective in patients with asthma who are also active smokers. In these patients, ICS treatment stimulates fewer short-term lung function improvements and anti-inflammatory effects compared to ex-smokers [16,17]. Thus, smoking can contribute to a reduced effect of systemic corticosteroids in COPD patients with acute exacerbations. 

The smoking status of participants in the CORTICO-COP trial was carefully documented. Therefore, the influence of smoking status on the effects of systemic corticosteroids in patients with severe COPD exacerbations could be analysed in this dataset while preserving the random allocation to corticosteroid-sparing regimen vs. a 5-day corticosteroid regimen. This post hoc subgroup analysis thus investigated whether eosinophil-guided corticosteroid-sparing treatment for acute exacerbations of COPD was non-inferior to conventional treatment in the two randomly allocated treatment arms, in current smokers and ex-smokers. 

## 2. Materials and Methods

### 2.1. Study Design

This was a post hoc, subgroup analysis of the CORTICO-COP trial, which recruited participants from 3 August 2016 until 8 January 2019. The primary endpoint was days alive and out of hospital within 14 days (DAOH14). The secondary endpoints were: (i) time to COPD exacerbation within 6 months, (ii) admission to intensive care unit (ICU) or death within 6 months, (iii) increase in lung function within 3 months, and (iv) health-related quality of life assessed by COPD Assessment Test (CAT) within 3 months. Data on time to COPD exacerbation within 6 months and admission to intensive care unit (ICU) or death within 6 months were obtained from the patient medical records and have not previously been reported in the CORTICO-COP trial.

### 2.2. Study Population

The study population consisted of patients from the CORTICO-COP trial, a nationwide multicentre prospective trial (*N* = 318), investigating eosinophil-guided corticosteroid treatment for hospitalisation-requiring acute exacerbation of COPD [12]. All consecutive patients admitted to the wards of the participating sites were eligible if they were included within 24 h of admission, were aged at least 40 years old, had known airflow limitation (defined as post-bronchodilator forced expiratory volume in 1 s (FEV_1_)/forced vital capacity ratio ≤ 0.70), and a specialist-verified diagnosis of COPD based on stable disease-state data. Exacerbations were defined according to the consensus definition described by the Global Initiative for Chronic Obstructive Lung Disease (GOLD) committee: acute worsening of respiratory symptoms that result in additional therapy. Exclusion criteria included self-reported or physician-diagnosed asthma, life expectancy of less than 30 days, severe COPD exacerbation requiring invasive ventilation or ICU admission, allergy to systemic corticosteroids, inability to provide written informed consent, pregnancy or lactation, systemic fungal infections, or patients receiving more than 10 mg of maintenance systemic corticosteroids daily. Written informed consent was obtained from patients before randomisation. Patients could only participate in the trial once. 

### 2.3. Statistical Analysis

Patients included in the present main analysis were smokers from the eosinophil-guided group or the control group, as assessed at baseline. A sensitivity analysis was performed on ex-smokers from the eosinophil-guided and control groups. Smoking status was defined as smokers (minimum 10 pack-years) and ex-smokers (former smokers with smoking cessation for more than 6 months prior to study inclusion). The baseline variables included age, sex, body mass index, cigarette pack-years, blood eosinophil count at baseline, FEV_1_, Medical Research Council Dyspnoea Scale, CAT, treatment at baseline (long-acting beta-2 agonists (LABAs), long-acting muscarinic antagonists, ICS, and LABA/ICS), and COPD exacerbation history. Categorical data were analysed using chi-square and Fisher’s exact tests. The primary outcome was evaluated using *t*-tests. The Cox proportional hazards model was used to determine the time to COPD exacerbation within 6 months. Admission to ICU or death within 6 months was calculated using logistic regression analysis. The changes from baseline in FEV_1_ at 3 months were analysed using *t*-tests. Statistical analyses were carried out using SAS software (ver. 9.4; SAS Institute, Inc., Cary, NC, USA) and R software (ver. 3.4.3; R Development Core Team, Vienna, Austria).

### 2.4. Sample Size

The sample size was fixed because 104 current smokers participated in the CORTICO-COP trial. Thus, we performed a power calculation for the *t*-test (assuming a normal distribution). The standard deviation for DAOH14 was approximately 3.8 days. We hypothesised that in the population of daily smokers randomised in the CORTICO-COP trial, the eosinophil-guided reduction in systemic corticosteroids of approximately 60%, published in the main analysis, would decrease DAOH14 by 2 days or more, as smokers may be less responsive to corticosteroids. Therefore, the power of the analysis (two-sided) was 0.75.

## 3. Results

Patients included in the CORTICO-COP trial were screened for eligibility and subsequently recruited for participation in the trial. Of these patients, 104 were active smokers, 208 were ex-smokers, and 6 were non-smokers. Data from these patients were included in the post hoc analyses (Figure 1). 

Of the 104 active smokers, 50 patients had been randomised to the control arm of the study, whereas 54 patients were assigned to the eosinophil-guided study arm. The median ages of these groups were 73 and 70 years, respectively, and the median number of pack-years for both groups was 50. Of the 208 ex-smokers, 105 were assigned to the control arm and 103 were assigned to the eosinophil-guided study arm. The median ages of these groups were 76 and 77 years, respectively, and the median number of pack-years was 45 and 40, respectively. Further baseline data describing the patients included in this analysis are shown in Table 1. The proportion of patients receiving systemic corticosteroid treatment was markedly reduced in the eosinophil-guided treatment regimen on days 2–5 in comparison to day 1 in both smokers and ex-smokers (Figure 2).

### 3.1. Primary Outcome

Assessment of the primary outcome showed no significant difference in the number of DAOH14 between the two study arms in smokers. The mean DAOH14 was 9.8 days (95% confidence interval (CI), 8.7–10.8) for the control group and 8.7 days (95% CI, 7.5–9.9) for the eosinophil-guided group (*p* = 0.34; Table 2).

### 3.2. Secondary Outcomes

When examining the two study groups in terms of the secondary outcomes, we found no significant differences in the number of exacerbations or death within 6 months (*n* = 30 vs. *n* = 31; hazard ratio, 0.99 (95% CI, 0.60–1.64); *p* = 0.98), the frequency of ICU admissions or deaths within 6 months (*n* = 18 vs. *n* = 20; odds ratio, 1.05 (95% CI, 0.47–2.33); *p* = 0.91), lung function (FEV_1_) improvement within 3 months (mean difference 8.0 (9.3) vs. 10.1 (13.0); *p* = 0.46), or change in health-related quality of life (CAT) within 3 months (mean difference −3.4 (9.4) vs. −4.5 (6.4); *p* = 0.56) Table 2). 

### 3.3. Sensitivity Analysis in Ex-Smokers

Of the 208 ex-smokers, 105 patients were randomised to the control arm of the study and 103 patients were assigned to the eosinophil-guided study arm. Among patients in the eosinophil-guided groups, the proportion of patients receiving systemic corticosteroid treatment was slightly higher for ex-smokers than for smokers (Figure 2). 

The ex-smokers in the two study arms exhibited no significant difference in DAOH14. The mean DAOH14 was 9.3 days (95% CI, 8.5–10.0) for the control group and 9.0 days (95% CI, 8.2–9.8) for the eosinophil-guided group (*p* = 0.61; Table 3).

Analysis of secondary outcomes also showed no significant differences in the number of exacerbations or death within 6 months (*n* = 51 vs. *n* = 53; hazard ratio, 1.17 (95% CI, 0.8–1.72); *p* = 0.43), the number of ICU admissions or deaths within 6 months (*n* = 41 vs. *n* = 44; odds ratio, 1.16 [95% CI, 0.62–1.64]; *p* = 0.59), lung function (FEV_1_) improvement within 3 months (mean difference (SD) 6.7 (12.4) vs. 5.1 (11.9); *p* = 0.48), or change in health-related quality of life (CAT) within 3 months (mean difference (SD) −3.4 (7.0) vs. −3.0 (6.4); *p* = 0.75; Table 3). Figure 3 shows time to exacerbation or death in the two study arms.

## 4. Discussion

Among smokers, we found that there was no significant difference in the primary outcome (DAOH14) between the control group and the eosinophil-guided corticosteroid-sparing treatment group in our study. Assessment of secondary outcomes also revealed no significant differences between the two study arms. Similar results were observed in our sensitivity analysis for ex-smokers. Our results indicate that eosinophil-guided corticosteroid treatment is equally non-inferior to the standard of care in current smokers and ex-smokers with regard to the primary and secondary outcomes. However, we noted a slightly higher exposure to systemic corticosteroids in ex-smokers. 

Not only do inflammation and oxidative stress from cigarette smoking contribute to the pathogenesis of COPD, but smoking also contributes to [18] and perpetuates [19] corticosteroid resistance in patients with COPD. In particular, ICS are significantly less effective in suppressing airway inflammation in current smokers than in ex-smokers [20,21]. Moreover, current smoking status influences the long-term effects of ICS on lung function decline [22]. Inhaled budesonide has no effect on lung function decline [23] or progression of emphysema [24] compared to the placebo in current smokers with COPD. Furthermore, ICS are less effective in reducing COPD exacerbations in current smokers than in ex-smokers [21]. A previous study found that smoking status was not associated with the effectiveness of systemic corticosteroids in reducing COPD exacerbations [25]. 

Several mechanisms have been proposed to explain the reduction in corticosteroid response identified in patients with COPD, including upregulation of pro-inflammatory cytokines as a result of increased lung inflammation, reduced activity of corticosteroid receptor (GR), reduced expression of histone deacetylase-2, altered expression of surfactant protein D, and loss of Mucin 1 cytoplasmic tail expression [26,27,28]. However, relevant research in this field is lacking, which creates a major barrier to effective management of COPD in patients with reduced responsiveness to the anti-inflammatory effects of corticosteroids. This emphasises the need for further research into the molecular mechanisms responsible for corticosteroid resistance and highlights the importance of personalisation and optimisation of corticosteroid treatment for patients with COPD to control symptoms.

Measuring blood eosinophils has clinical potential for tailoring systemic corticosteroid treatment for COPD acute exacerbations [14]. Previous trials have shown that eosinophil-guided systemic corticosteroid treatment for acute exacerbation of COPD was non-inferior to standard of care [12,13]. However, the impact of smoking status on the efficacy of this treatment has not been well-characterised. A meta-analysis of eosinophil-guided prednisolone therapy by Bafadhel et al. concluded that smoking history did not affect the overall results, but current smoking status was been analysed [29]. Furthermore, most studies investigating the relationship between smoking/lung inflammation and corticosteroid resistance focused on ICS and not systemic corticosteroids. 

Previous analyses on the ECLIPSE study showed that the proportion of current smokers was lower among patients with non-exacerbated COPD and high eosinophil counts (>2%) than among those who did not have eosinophilia [30]. However, as suggested by previous studies [31,32], current smoking was not associated with lower blood eosinophil counts on the first day of hospitalisation. These results suggest that smoking status does not determine whether exacerbations are associated with eosinophilia. Moreover, corticosteroid treatment significantly reduced blood eosinophil counts in both current and ex-smokers, suggesting that smoking status does not influence the role of eosinophils as a biomarker for treating COPD exacerbations.

The proportion of patients exhibiting exacerbations or mortality at 6 months was higher among current smokers (59%) than among ex-smokers (50%). This observation is consistent with results from a previous study, which showed that current smoking status was a better predictor of a shorter time to the next exacerbation than the eosinophil number at the index event [31]. One previous study found that smoking status was not associated with a higher risk of readmission; however, this conclusion may be attributable to the low number of subjects included [33]. We did not investigate this aspect in detail, but smoking may hamper the maintenance treatment that is designed to prevent COPD exacerbation. Notably, these analyses were been adjusted for potential confounders, such as age, lung function, medications, or comorbidities.

The strengths of this study include the randomised study population and complete follow-up of the primary and secondary outcomes. In addition, all patients in the study had a COPD diagnosis that was confirmed by a respiratory medicine specialist and validated at least once a year, as well as a smoking status that was registered at every outpatient visit, reducing the risk of misclassification bias.

This study also had some limitations. First, this post hoc analysis was limited by the retrospective nature of data collection. Second, the small sample size may have been insufficient to power the statistical analysis. Third, our analyses were not adjusted to account for known confounders, which is a potential source of error.

## 5. Conclusions

Our results suggest that eosinophil-guided corticosteroid-sparing treatment is equally non-inferior to the standard of care in current smokers and ex-smokers (sensitivity analysis) with regard to primary and secondary outcomes. Because little is known about the impact of smoking status on eosinophil-guided corticosteroid-sparing treatment, more randomised trials are needed to confirm our results. Although smoking status did not influence the role of eosinophils as a biomarker in guiding corticosteroid treatment, current smokers are at higher risk of readmission; therefore, smoking cessation should be considered for all patients admitted with COPD exacerbations.

## Figures and Tables

**Figure 1 jcm-10-02734-f001:**
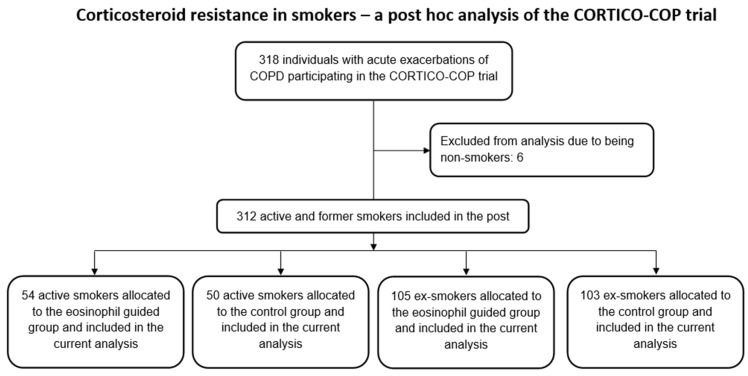
Study flowchart.

**Figure 2 jcm-10-02734-f002:**
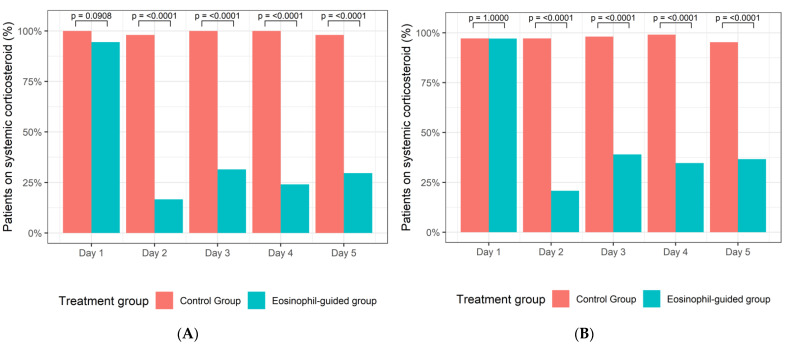
(**A**) Percentage of smokers in the control group and the eosinophil-guided group receiving systemic corticosteroids from days 1–5. (**B**) Percentage of ex-smokers in the control group and the eosinophil-guided group receiving systemic corticosteroids from days 1–5.

**Figure 3 jcm-10-02734-f003:**
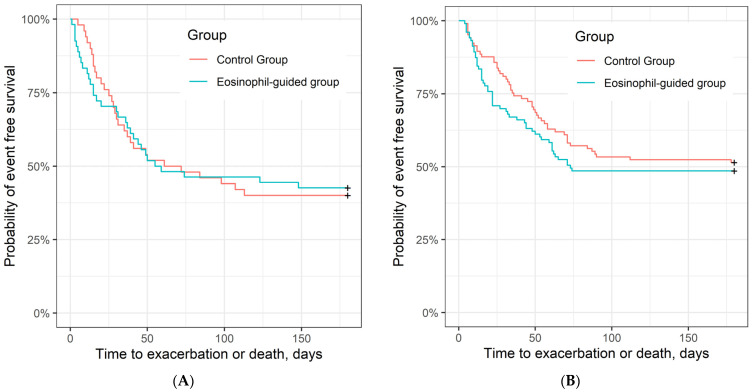
Time to exacerbation or death (unadjusted). (**A**) Kaplan–Meier curve of time to exacerbation or death in control group and eosinophil-guided group in current smokers, HR = 0.99 (95% CI, 0.60–1.64), *p* = 0.98 (log-rank test) and (**B**) in ex-smokers, HR = 1.17 (95% CI, 0.8–1.72); *p* = 0.43.

**Table 1 jcm-10-02734-t001:** Baseline characteristics of current and ex-smokers in the CORTICO-COP trial.

	All Current Smokers	Current Smokers Control Group	Current Smokers Eosinophil-Guided Group	All Ex-Smokers	Ex-Smokers Control Group	Ex-Smokers Eosinophil-Guided Group
Number, *n* (%)	104 (100)	50 (100)	54 (100)	208 (100)	105 (100)	103 (100)
Age, median (IQR), years	72 (66–78)	73 (66–78)	70 (66–77)	76 (71–83)	76 (71–83)	77 (72–83)
Male sex, *n* (%)	47 (45)	20 (40)	27 (50)	94 (45)	48 (46)	46 (45)
BMI, median (IQR), kg/m^2^	23.2 (20.0–27.7)	21.9 (19.3–27.5)	23.7 (20.6–27.5)	24.2 (20,8–27.4)	24.4 (20.8–27.9)	24.2 (20.8–26.3)
Pack-years history of current or prior smoker (median, IQR), years	50 (40–60)	50 (40–60)	50 (40–60)	40 (30–50)	45 (30–50)	40 (28–50)
Blood eosinophil count, median (IQR), ×10^9^ cells/L	0.09 (0.01–0.20)	0.07 (0.01–0.15)	0.10 (0.01–0.29)	0.07 (0.01–0.27)	0.06 (0.01–0.23)	0.10 (0.01–0.30)
FEV_1_, median (IQR), %	32 (23–40)	30 (22–40)	32 (25–39)	30 (23–39)	30 (24–40)	30 (23–38)
CAT scores	23 (18–27)	21 (18–27)	23 (18–28)	21 (16–25)	21 (14–25)	21 (17–25)
MRC, median (IQR)	4 (3–5)	4 (3–5)	4 (3–5)	4 (3–5)	4 (3–4)	4 (3–5)
MRC 1, *n* (%)	3 (3)	0 (0)	3 (6)	5 (2)	3 (3)	2 (2)
MRC 2, *n* (%)	9 (9)	4 (8)	5 (10)	14 (7)	5 (5)	9 (9)
MRC 3, *n* (%)	26 (25)	16 (32)	10 (19)	64 (31)	34 (33)	30 (30)
MRC 4, *n* (%)	30 (29)	11 (22)	19 (37)	69 (34)	39 (38)	30 (30)
MRC 5, *n* (%)	34 (33)	19 (38)	15 (29)	52 (25)	23 (22)	29 (29)
Treatment with LABA alone, *n* (%)	19 (18)	7 (14)	12 (22)	65 (31)	26 (25)	39 (38)
Treatment with LAMA alone, *n* (%)	9 (9)	4 (8)	5 (9)	20 (10)	15 (14)	5 (5)
Treatment with ICS alone, *n* (%)	5 (5)	2 (4)	3 (6)	4 (2)	1 (1)	3 (3)
Treatment with ICS/LABA, *n* (%)	58 (56)	31 (62)	27 (50)	106 (51)	60 (57)	46 (45)
None of the above, *n* (%)	13 (12)	6 (12)	7 (13)	13 (6)	3 (3)	10 (10)
0 severe exacerbations 12 months prior to baseline, *n* (%)	72 (69)	35 (70)	37 (69)	136 (65)	68 (65)	68 (66)
1 severe exacerbation 12 months prior to baseline, *n* (%)	17 (16)	8 (16)	9 (17)	41 (20)	20 (19)	21 (20)
≥2 severe exacerbations 12 months prior to baseline, *n* (%)	15 (14)	7 (14)	8 (15)	31 (15)	17 (16)	14 (14)
Ischaemic heart disease, *n* (%)	15 (14)	8 (16)	7 (13)	21 (10)	6 (6)	15 (15)
Hypertension, *n* (%)	39 (38)	18 (36)	21 (39)	82 (39)	41 (39)	41 (40)
Hypercholesterolemia, *n* (%)	15 (14)	10 (20)	5 (9)	22 (11)	9 (9)	13 (13)
Chronic renal failure, *n* (%)	2 (2)	2 (4)	0 (0)	19 (9)	7 (7)	12 (12)
Heart failure, *n* (%)	6 (6)	4 (8)	2 (4)	24 (12)	9 (9)	15 (15)
Osteoporosis, *n* (%)	18 (17)	9 (18)	9 (17)	39 (19)	15 (14)	24 (23)

Abbreviations: BMI, body mass index; FEV_1_, forced expiratory volume in 1 s; ICS, inhaled corticosteroids; IQR, interquartile range; LABAs, long-acting beta-2 agonists; LAMAs, long-acting muscarinic antagonists; MRC, Medical Research Council Dyspnoea Scale.

**Table 2 jcm-10-02734-t002:** Main analysis on current smokers.

	All *n* = 104	Control Group *n* = 54	Eosinophil-Guided Group *n* = 50	*p*-Value
DAOH14, mean (95% CI)	9.2 (4.1)	9.8 (8.7–10.8)	8.7 (7.5–9.9)	0.34
Exacerbation or death within 6 months, *n* (%)	61 (59)	30 (60)	31 (57)	
HR (95% CI)		Reference	0.99 (0.60–1.64)	0.98
ICU or death within 6 months, *n* (%)	38 (37)	18 (36)	20 (37)	
OR (95% CI)		Reference	1.05 (0.47–2.33)	0.91
FEV_1_ pct				
Baseline, mean (SD)	33.3 (13,1)	31.7 (12.3)	34.7 (13.7)	
Day 90, mean (SD)	42.2(15.7)	40.0 (15.6)	44.3 (15.7)	
Difference, mean (SD)	9.1 (11.3)	8.0 (9.3)	10.1 (13.0)	0.46
CAT score				
Baseline, mean (SD)	22.3 (7.6)	22.2 (7.3)	22.5 (7.9)	
Day 90, mean (SD)	18.0 (7.8)	19.2 (7.0)	16.8 (8.4)	
Difference, mean (SD)	−4.0 (8.0)	−3.4 (9.4)	−4.5 (6.4)	0.56

Abbreviations: DAOH14, days alive and out of hospital within 14 days; CI, confidence interval; CAT, COPD Assessment Test; FEV_1_, forced expiratory volume in 1 s; HR, hazard ratio; OR, odds ratio; ICU, intensive care unit; SD, standard deviation.

**Table 3 jcm-10-02734-t003:** Sensitivity analysis of ex-smokers.

	All *n* = 208	Control Group *n* = 105	Eosinophil-Guided Group *n* = 103	*p*-Value
DAOH14, mean (95% CI)	9.1 (8.6–9.7)	9.3 (8.5–10.0)	9.0 (8.2–9.8)	0.61
Exacerbation or death within 6 months, *n* (%)	104 (50)	51 (49)	53 (51)	
HR (95% CI)		Reference	1.17 (0.80–1.72)	0.43
ICU or death within 6 months, *n* (%)	85 (41)	41 (39)	44 (43)	
OR (95% CI)		Reference	1.16 (0.62–1.64)	0.59
FEV_1_ pct.				
Baseline, mean (SD)	32.5 (13.2)	32.3 (12.9)	32.7 (13.5)	
Day 90, mean (SD)	41.5 (17.5)	41.2 (15.9)	41.8 (19.3)	
Difference, mean (SD)	6.0 (12.1)	6.7 (12.4)	5.1 (11.9)	0.48
CAT score				
Baseline, mean (SD)	20.6 (7.2)	20.3 (7.6)	20.9 (6.7)	
Day 90, mean (SD)	17.5 (7.1)	17.1 (8.0)	18.0 (5.6)	
Difference, mean (SD)	−3.2 (6.7)	−3.4 (7.0)	−3.0 (6.4)	0.75

Abbreviations: CI, confidence interval; DAOH14, days alive and out of hospital within 14 days; CAT, COPD Assessment Test; FEV_1_, forced expiratory volume in 1 s; HR, hazard ratio; OR, odds ratio; ICU, intensive care unit; SD, standard deviation.

## Data Availability

We think that knowledge sharing increases the quantity and quality of scientific results. Sharing of relevant data will be discussed within the study group upon reasonable request.

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
