# Peer review of "Corticosteroid Resistance in Smokers—A Substudy Analysis of the CORTICO-COP Randomised Controlled Trial"

_jcm, 2021, doi:10.3390/jcm10122734_

Round 1

Reviewer 1 Report

Sivapalan et al report a post-hoc subgroup analysis of the CORTICO-COP RCT trial, which previously demonstrated non-inferiority in clinical outcomes following exacerbations of COPD comparing a blood-eosinophil guided OCS treatment intervention vs standard treatment intervention. The clinical benefit of the eosinophil-guided intervention related to reduced OCS burden/use compared to standard treatment regimens. The present analysis compares clinical outcomes of these two intervention arms in two sub-groups, current smokers, and ex-smokers. The presented study tests a sound hypothesis with valid methods in the well-described CORTICO-COP study population, and provides important additional conclusions relating to the non-inferiority of the OCS-sparing eosinophil-guided treatment approach irrespective of smoker/ex-smoker status in individuals exacerbating with COPD. The analysis methodology seems appropriate, and the main limitations of the study are appropriately acknowledged by the authors in the discussion. I have some minor specific comments:

Methods section 2.1: The 6 month time frame for some of the secondary endpoints differs from that reported in the original cortico-cop trial (Sivapalan et al Lancet Resp Med 2019), where time to COPD exacerbation and admission to ICU or death were evaluated over 30 days post baseline. Can the authors briefly expand in the method as to how this extra data was captured for the present study?

Methods section 2.4: Sample size calculation lines 122-125 “We hypothesised that in the population of daily smokers randomised in the CORTICO-COP trial, the eosinophil-guided reduction in systemic corticosteroids of approximately 60%, published in the main analysis, would decrease DAOH14 by 2 days or more.” Not sure I understand the basis of this hypothesis. Original trial showed no difference in DAOH14 between eosinophil guided and standard treatment regimens.

Table 2: CAT score changes. Is difference, mean (SD) displayed correctly? unsure why mean difference is -4 in All, vs +9.4 and +6.4 in control and eo guided groups respectively. I believe this is a typo, particularly when looking at subsequent Table 3.

Figure 3: what is the p value in figure legend Fig 3 referring to - panel A or B? the p value is not described in the results text. What is the result of log rank test for each of the two panels A and B? It would be informative to see both sets of data (smokers, ex-smokers) tested. Please clarify.

Discussion paragraph 1 lines 201-202: “However, we noted a slightly lower exposure to systemic corticosteroids by ex-smokers.” this conclusion statement seems contradictory to prior statements in the results, and what is displayed in the figure, where it appears slightly higher proportions of ex-smokers in the eosinophil guided group were prescribed OCS over the course of the intervention compared to current smokers.

Discussion: discussion lines 232-236 seem a bit repetitive from previous discussion

Author Response

We would like to thank the reviewers and editors for valuable comments. We have revised the paper according to these comments, and we believe it has been improved. Below are our point-by-point responses to the comments made by the reviewers. The reviewers’ comments are listed ‘C1, C2, C3, etc.’ and our responses are marked ‘R_C1, R_C2, etc.’

We believe the revised manuscript has improved substantially the reporting of the study.

Pradeesh Sivapalan MD PhD, Andras Bikov MD PhD and Professor Jens-Ulrik Stæhr Jensen MD PhD

University of Copenhagen and Herlev-Gentofte Hospital

Respiratory Research Section, Department of Medicine, Herlev-Gentofte Hospital

Comments from the reviewers:

Reviewer: 1

Comment to the authors:

C1: Methods section 2.1: The 6 month time frame for some of the secondary endpoints differs from that reported in the original cortico-cop trial (Sivapalan et al Lancet Resp Med 2019), where time to COPD exacerbation and admission to ICU or death were evaluated over 30 days post baseline. Can the authors briefly expand in the method as to how this extra data was captured for the present study?

R_C1: Thank you for making us aware of this. It is correct that information on time to COPD exacerbation within 6 months and admission to intensive care unit (ICU) or death within 6 months had not been reported in the CORTICO-COP trial. This information was retrieved from the patient medical records. This has now been clarified in the manuscript.

Page 2 line 85 – 92: “Data on time to COPD exacerbation within 6 months and admission to intensive care unit (ICU) or death within 6 months was obtained from the patient medical records and has not previously been reported in the CORTICO-COP trial.”

C2: Methods section 2.4: Sample size calculation lines 122-125 “We hypothesised that in the population of daily smokers randomised in the CORTICO-COP trial, the eosinophil-guided reduction in systemic corticosteroids of approximately 60%, published in the main analysis, would decrease DAOH14 by 2 days or more.” Not sure I understand the basis of this hypothesis. Original trial showed no difference in DAOH14 between eosinophil guided and standard treatment regimens.

R_C2: Thank you for this comment. Low DAOH14 corresponds to more days spent in hospital or dead. Smokers may be less responsive to corticosteroids, so reducing these drugs among smokers may more often lead to insufficient dosing and thereby a reduced effect.

We did not explain this clear enough in the rationale – this has now been updated:

Page 4 line 125 – 129: “We hypothesised that in the population of daily smokers randomised in the CORTICO-COP trial, the eosinophil-guided reduction in systemic corticosteroids of approximately 60%, published in the main analysis, would decrease DAOH14 by 2 days or more, as smokers may be less responsive to corticosteroids.”

C3: Table 2: CAT score changes. Is difference, mean (SD) displayed correctly? unsure why mean difference is -4 in All, vs +9.4 and +6.4 in control and eo guided groups respectively. I believe this is a typo, particularly when looking at subsequent Table 3.

R_C3: Thank you very much for making us aware of this. It was a typographical error. This has now been changed in the table and in the results section.

Page 7 table 2, last row.

Difference, mean (SD)

-4.0 (8.0)

-3.4 (9.4)

-4.5 (6.4)

0.56

Page 6 line 171 – 172: “When examining the two study groups……………..or change in health-related quality of life (CAT) within 3 months (mean difference -3.4 (9.4) vs. -4.5 (6.4); p = 0.56) Table 2).”

C4: Figure 3: what is the p value in figure legend Fig 3 referring to - panel A or B? the p value is not described in the results text. What is the result of log rank test for each of the two panels A and B? It would be informative to see both sets of data (smokers, ex-smokers) tested. Please clarify.

R_C4: Thank you for this comment. We have now added the hazard ratios and results of log rank test for each of the two panels (A and B).

Page 7 figure legend to figure 3, line 178 - 180. “Time to exacerbation or death (unadjusted). (A) Kaplan–Meier curve of time to exacerbation or death in control group and eosinophil-guided group in current smokers, HR = 0.99 [95% CI, 0.60–1.64], p = 0.98 (log rank test) and in (B) ex-smokers, HR = 1.17 [95% CI, 0.8–1.72]; p = 0.43”

C5: Discussion paragraph 1 lines 201-202: “However, we noted a slightly lower exposure to systemic corticosteroids by ex-smokers.” this conclusion statement seems contradictory to prior statements in the results, and what is displayed in the figure, where it appears slightly higher proportions of ex-smokers in the eosinophil guided group were prescribed OCS over the course of the intervention compared to current smokers.

R_C5: Thank you for this comment. We apologize. This was a mistake. This has been changed to "However, we noted a slightly higher exposure to systemic corticosteroids by ex-smokers." Which also corresponds to Figure 2

C6: Discussion: discussion lines 232-236 seem a bit repetitive from previous discussion

R_C6: Thank you for this comment. We agree with you. We have deleted this part: “To our knowledge, this is the first study to explore the effect of current smoking status on eosinophil-guided systemic corticosteroid-sparing treatment in randomised study arms. The results from this study suggest that in current smokers, eosinophil-guided corticosteroid-sparing treatment for acute exacerbation of COPD is non-inferior to conventional treatment

Reviewer 2 Report

This article is a sub-study of the CORTICO-COP study, a multi-center, randomized, controlled, open-label trial designed to test if standard care compared to eosinophil-guided systemic corticosteroids -sparing therapy is non-inferior regarding length of hospital stay. Since smoking status has been shown to affect corticosteroid responsiveness, the authors designed a study to assess whether eosinophil-guided treatment was non-inferior to conventional treatment in current smokers.

The study is very methodologically well-designed to answer the question and although the number of patients is limited, it is high enough to get additional information although new CT including a higher number of patients are needed. The article is well-written and the results and conclusions contribute new knowledge to the field and are clinically relevant.

Author Response

Reviewer: 2

Comment to the authors:

C1: This article is a sub-study of the CORTICO-COP study, a multi-center, randomized, controlled, open-label trial designed to test if standard care compared to eosinophil-guided systemic corticosteroids-sparing therapy is non-inferior regarding length of hospital stay. Since smoking status has been shown to affect corticosteroid responsiveness, the authors designed a study to assess whether eosinophil-guided treatment was non-inferior to conventional treatment in current smokers.

The study is very methodologically well-designed to answer the question and although the number of patients is limited, it is high enough to get additional information although new CT including a higher number of patients are needed. The article is well-written and the results and conclusions contribute new knowledge to the field and are clinically relevant.

R_C1: Thank you very much for this comment. We are happy to contribute to new knowledge. We hope that our results can contribute to future research questions in randomized controlled trials.